# A Systematic Review: Is Porcine Kobuvirus Causing Gastrointestinal Disease in Young Pigs?

**DOI:** 10.3390/vetsci10040286

**Published:** 2023-04-11

**Authors:** Esben Østergaard Eriksen

**Affiliations:** Section for Production, Nutrition and Health, Department of Veterinary and Animal Sciences, Faculty of Health and Medical Sciences, University of Copenhagen, 1870 Frederiksberg C, Denmark; esbene@sund.ku.dk

**Keywords:** kobuvirus, aichivirus, picornaviridae, pig, gastrointestinal disease, diarrhea

## Abstract

**Simple Summary:**

In 2008, researchers described a new virus, porcine kobuvirus (PKV). Some viruses are very harmful, while others infect animals without having any significance for their health. Some researchers have suspected that porcine kobuvirus causes gastrointestinal disease (e.g., diarrhea) in pigs. Therefore, this paper tried to answer the question: Is porcine kobuvirus a cause of gastrointestinal disease in young pigs? A systematic literature review was conducted, meaning that a database was searched for all reports of research studies investigating kobuvirus in pigs. In general, there was not much research of good quality that could possibly answer the question. Therefore, the study concluded that there is a lack of good evidence supporting the idea that PKV causes gastrointestinal disease. The absence of such documentation does not mean that we can conclude the opposite: that PKV is not causing gastrointestinal disease. That said, the sparse research available did indicate that PKV has a limited ability to cause diarrhea.

**Abstract:**

Since porcine kobuvirus (PKV) was first described in 2008, researchers have speculated whether the virus is of clinical importance. This systematic literature review answers the question: Is porcine kobuvirus a cause of gastrointestinal disease in young pigs? A case-control study showed that PKV was not associated with neonatal diarrhea. A cohort study suffered from a very small sample size (*n* = 5), and in an experimental trial, the effect of PKV inoculation could not be separated from the effect of being inoculated with porcine epidemic diarrhea virus. In 13 poorly defined observational studies, more than 4000 young pigs had been assigned a diarrhea status and their feces analyzed for PKV. Unfortunately, the studies lacked well-characterized unbiased samples, and thus the strongest possible inference from these studies was that a very strong association between PKV and diarrhea is unlikely. PKV was commonly detected in non-diarrheic pigs, and this could indicate that PKV is not a sufficient cause in itself or that reinfection of individuals with some immunological protection due to previous infections is common. Conclusively, there is a lack of good evidence of PKV being a cause of gastrointestinal disease, but the sparse available evidence suggests that PKV is of limited clinical importance.

## 1. Introduction

Kobuvirus is a non-enveloped single-stranded positive-sense genomic RNA virus and a member of the *Picornaviridae* family [1]. Kobuvirus was first isolated in 1989 from humans suffering from acute gastroenteritis associated with oyster consumption [2]. Since then, kobuviruses have been discovered in domestic animals, including cats [3], cattle [4], dogs [5], goats [6], sheep [7], and pigs. Porcine kobuvirus (PKV) was first detected in 2007 in normal feces from neonatal, healthy piglets living in a Hungarian pig herd [8]. The virus appears to be present worldwide; so far, PKV has been reported in Austria [9], Brazil [10], Canada [11], China [12], the Czech Republic [13], Belgium [14], Denmark [15], France [16], Germany [9], Greece [17], Hungary [8], India [18], Ireland [19], Italy [20], Japan [21], Kenya [22], Korea [23], Mexico [24], the Netherlands [10], Serbia [25], Slovakia [26], Spain [9], Sweden [9], Thailand [27], Uganda [22], the United States of America [28], and Vietnam [29].

The PKV virion has a diameter of ~30 nm, and the genome consists of approximately 8100–8200 nucleotides [30]. Evidence from other species suggests that transmission occurs through the fecal-oral route [31,32]. The virus is able to infect the cells of the villi in the small intestine as well as lymphocytes in Peyer’s patches [33]. A mechanism by which PKV might overcome the first innate immune response to viral infections, the interferon system, has been suggested. The virus protein 3 might inhibit the interferon-β pathway, as it could be able to block the transcription of genes in the cell nucleus that is normally induced by this signal molecule [34]. After infection, pigs seem to develop immunity against PKV for some time [35].

The pathogenic capabilities of PKV were questioned in the first publication describing the virus in 2008 [8]. In 2009, after detecting PKV at a very high frequency (99%, *n* = 97/98) in diarrheic pigs in Thailand, researchers concluded that studies clarifying the role of PKV in gastrointestinal disease were warranted [27]. Half a year later, a group of researchers suspected that PKV could be an intestinal pathogen and called for further research [23]. Since then, multiple studies have been conducted. Yet, it is unclear whether PKV can be considered a cause of gastrointestinal disease in pigs, and no reviews of the published literature were available. A recent review on the infectious causes of neonatal piglet diarrhea briefly mentioned PKV and other newly discovered viruses and stated that their involvement in disease “*is not well defined*” [36]. Therefore, the objective of this paper was to review the evidence on porcine kobuvirus as a cause of gastrointestinal disease in young pigs.

## 2. Materials and Methods

A systematic review was conducted. For this purpose, a research question was formulated based on the PICO model:

In young pigs, does infection with porcine kobuvirus increase the risk of gastrointestinal disease?

The population (P), exposure/intervention (I), comparison (C), and outcome (O) are more precisely defined in Table 1. The target population, for which external validity was sought, was young pigs living in commercial, intensified indoor pig productions; however, studies (e.g., experimental trials) housing pigs in other environments (e.g., research facilities) producing results that could possibly be extrapolated to this target population were also considered for inclusion in the review.

Searches were performed in the Web of Science database. Two blocks representing the population and exposure (Table 1) were searched for using the Web of Science topic indexing:

TOPIC: (kobuvirus OR “aichivirus c”) AND TOPIC: (pig* OR swine* OR porcine).

The first search was performed on 11 February 2020, and a final search was performed again on 19 February 2023. The review included all peer-reviewed original research publications in the English language reporting research with any type of study design. Studies were excluded if they did not provide empirical evidence that was able to answer the PICO-based question (Table 1). Data were extracted from a set of observational studies. Whenever possible, the data was collected stratified by age groups, and when it was not reported in the original paper, the percentage of PKV infection detected among both diarrheic and non-diarrheic pigs was estimated. Whenever possible, only data from young pigs were included from studies that also included data on mature pigs, not meeting the population description (Table 1).

## 3. Results

The final database search yielded 110 publications. Seventeen (17) papers reporting 16 studies included data that could possibly provide some clarity to the proposed question (Table 1). An overview of the included studies is presented in Table 2. Unsurprisingly, all studies have been conducted since the first description of PKV in December 2008 [8].

Studies that were not included were, for instance, some observational field studies establishing the occurrence of PKV, but only in individuals with diarrhea (e.g., [27]) or without diarrhea (e.g., [21]). One study was excluded because only four non-diarrheic pigs were tested [37]. Another study reported the presence of PKV in pigs with diarrhea and non-diarrheic pigs [38], but the age of all non-diarrheic pigs was above six months, i.e., too old to match the defined population (Table 1).

In general, the included literature was dominated by observational studies detecting PKV by real-time polymerase chain reaction (RT-PCR) in fecal samples collected in multiple farms at one time-point from pigs of different ages. These studies are summarized in Table 3, and they shared the characteristic that they had poorly defined study designs and sampling procedures. This imposed great limitations on the interpretations of the presented data, and it is questionable whether they actually met the inclusion criteria (see Discussion, Section 4.1). Nevertheless, the occurrence of PKV was often compared between the groups (those with diarrhea and those without diarrhea) in the studies. Finally, a well-designed case-control study [15,39], a cohort study with a very small sample size (*n* = 5) [14], and an experimental trial were included [40].
vetsci-10-00286-t002_Table 2Table 2Overview of 16 studies included in the systematic review of whether porcine kobuvirus is a cause of gastrointestinal disease in pigs.ReferenceCountryStudy Type*n*ExposureOutcome(s) [41]KoreaObservational *119PKV, fecal sample, RT-PCRDiarrhea † [42]Sichuan Province,  ChinaObservational *140PKV, fecal sample, RT-PCRDiarrhea † [43]ThailandObservational *638PKV, fecal sample, RT-PCRDiarrhea † [44]Shanghai Province, ChinaObservational *135PKV, fecal or intestinal sample, RT-PCRDiarrhea † [15]DenmarkCase-control96PKV, mix of ilium content and tissue, RT-PCR  (incl. viral load)Diarrhea ‡ [26]SlovakiaObservational *414PKV, fecal sample, RT-PCRDiarrhea † [11,35]CanadaObservational *130PKV, fecal sample, RT-PCRDiarrhea † [23]KoreaObservational *134PKV, fecal sample, RT-PCRDiarrhea † [40]Xinjiang Province, ChinaExperimental trial2 × 5Fecal sample inoculum containing PKV and porcine epidemic diarrhea virus vs. mock inoculumPKV by RT-PCR, clinical signs, rectal temperature, body weight, intestinal histopathology, and immunohistochemistry [14]BelgiumCohort5PKV, fecal sample, RT-PCRDiarrhea † [45]HungaryObservational *306 #PKV, fecal sample, RT-PCRDiarrhea † [29]Dong Thap Province, VietnamObservational *393PKV, fecal sample, RT-PCR (incl. viral load)Diarrhea † [46]Shanghai Province, ChinaObservational *116PKV, fecal sample, RT-PCRDiarrhea † [47]SlovakiaObservational *20 PKV, fecal sample, next generation sequencing techniqueDiarrhea † [48]United States of AmericaObservational *160PKV, fecal sample, RT-PCRDiarrhea † [49]ChinaObservational *1309PKV, fecal sample, RT-PCR (incl. viral load)Diarrhea † [9]Austria (*n* = 136), Germany (*n* = 44), Hungary (*n* = 50), Spain (*n* = 83), and Sweden (*n* = 106) $Observational *419PKV, fecal sample, RT-PCR (incl. viral load)Diarrhea †Studies are listed in alphabetic order according to the first author of the publications. * These studies were analyzing samples collected from observational field studies and laboratory submissions, but they all had poorly defined study designs, at least in the part of the investigation that was relevant for this review. † Diagnostic criteria for the assessment of diarrhea were not reported or defined. ‡ Diarrhea was defined as a fluid appearance on a rectal swab in two subsequent days; age-matched controls had normal feces when collected on a rectal swab daily from birth until inclusion at three to seven days of age [15,39]. # Data from sows was included in this review as they could not be separated from the young animals in the reported results. $ All samples from Germany were from diarrheic pigs.


## 4. Discussion

In this discussion paragraph, the included literature will first be critically reviewed, and this will be wrapped up with a conclusion according to the research question of this review: whether PKV is a cause of gastrointestinal disease in young pigs. The two final sub-paragraphs offer the author’s reflections and recommendations for future research and the practical implications for veterinary pig practitioners, respectively.

### 4.1. Poorly Defined Observational Studies

The occurrence of PKV is affected by factors such as the age of the pigs and the farm [16,35]. The poorly defined observational studies listed in Table 3 recruited pigs from different age groups and farms, generally described their sampling technique poorly, and never explicitly described how their sampling schemes sought to obtain a representative sample of pigs. Therefore, major biases were introduced to the studies in Table 3 by ignoring obvious confounders, such as the farm and the age of the pigs. For instance, in an American study, diarrheic samples submitted to a laboratory from farms located in 15 different states were compared to non-diarrheic samples collected from three farms in Minnesota [48]. Available descriptive statistics of the studied population in another study [44] also illustrate the problem well: Weaned pigs (*n* = 52) were proportionally over-represented in the non-diarrheic group (50% of all non-diarrheic pigs) compared to the diarrheic group (33% of all diarrheic pigs) [44].

Despite the poorly defined study designs, some studies even forced statistical tests on the data (e.g., [11,23,35,46]). The study that sparked the speculation of PKV’s pathogenic capabilities applied Pearson’s chi-square test, and it was concluded that PKV was present at a statistically significant higher frequency in diarrheic pigs [23]. The study did not recruit any diarrheic pigs from 11 out of the 40 included farms [23]. Thus, the pigs did not only differ in clinical status but also in herd of origin, and hence, the basic assumptions of random sampling and independence between subjects were violated. Therefore, the conclusion is not sound.

The poorly defined observational study from Vietnam [29] claimed random selection of the non-diarrheic pigs, yet the exact procedure for the random sampling was not specified. The inclusion of diarrheic pigs appeared to be through haphazard sampling. The issue was further complicated by the fact that only a subsample of the specimens were analyzed for PKV. The criteria for this subsample selection were not specified, though some matching on herd appeared to be intended (three to five non-diarrheic samples per herd were included) [29]. Another study, an observational study from Canada, followed pigs over time to establish the shedding of enteric viruses throughout the pigs’ lives [11,35]. The investigation recorded the diarrhea status of the pigs at one point in time; piglets less than 3 weeks of age were included based on their diarrhea status: Group 1, diarrheic pigs; Group 2, non-diarrheic pigs within a pen with diarrheic pen-mates; Group 3, non-diarrheic pigs within a pen without diarrhea. The design of this study appeared slightly stronger than the rest of the studies listed in Table 3, as it aimed for some sort of matching on the confounding effects of age, farm, and pen/sow. However, it was apparent from the distribution between groups (Gr. 1, *n* = 61; Gr. 2, *n* = 23; and Gr. 3, *n* = 97) sampled from 11 herds that exact herd and pen-wise matching have not been applied. The paper stated that the included pigs were identified by a veterinarian but did not report the use of random sampling to select the pigs. Conclusively, as for the rest of the studies listed in Table 3, the obtainment of unbiased, representative samples may not be assumed for the two above-mentioned studies, either.

Despite the poorly defined study designs, very strong associations could potentially have been obvious from the data. This encourages comparing the statistics with great caution. Before proceeding with this, it must be noted that strong associations do not necessarily imply a causal relation; strong associations may as well arise purely from confounding bias or collider bias [50]. As seen from Table 3, there was no clear tendency: Five studies detected PKV at a higher frequency in pigs with diarrhea [11,23,35,42,44,46]. Four studies found a somewhat similar risk of PKV infection in pigs with diarrhea and those without diarrhea [9,26,29,43]. Three studies detected PKV at a higher frequency in pigs with healthy feces [41,45,49]. If proceeding even further down the path of reckless comparisons by analyzing all the included data collectively, the occurrence of PKV was similar for diarrheic (43.7%, *n* = 1133/2590) and non-diarrheic pigs (41.1%, *n* = 675/1641). Apart from the results presented in Table 3 (PKV occurrence in diarrheic and non-diarrheic pigs), three of the studies [9,29,49] also presented quantitative measures of the viral load in the fecal samples. The analysis of a larger number of samples from Chinese pig productions showed similar PKV loads in diarrheic and non-diarrheic pigs in all four included age groups (<1 week, 1–3 weeks, 3–10 weeks, and 10–24 weeks). In a Vietnamese study, viral load comparisons between diarrheic and non-diarrheic pigs were only reported for mature animals (boars) and not the young animals of interest for this review [29]. In a study from five European countries [9], comparisons of Cq values were made between healthy and diarrheic fecal samples across all age groups (suckling pigs, weaners, growers), but separated by the country of origin (Austria, Hungary, Spain, Sweden). Results from this study were mixed, but generally not indicative of a difference in viral load between diarrheic and non-diarrheic fecal samples.

In summary, little inference may be drawn from the occurrences of PKV in diarrheic and non-diarrheic pigs presented in Table 3. At best, the presented data may indicate that a very strong association between PKV and prevalent diarrhea is unlikely and that PKV is a common finding in both healthy and diarrheic feces of young pigs.

Along with the criticism presented above, readers should be aware that many of the reviewed studies were of an explorative nature and had other main objectives than assessing the association between diarrhea and PKV (e.g., to develop laboratory diagnostics or clarify the genetics of PKV). None of the studies were labeled inappropriately as a case-control study or a cross-sectional study.

### 4.2. Other Studies

Diarrhea status was an outcome variable in all of the included studies, and the clinical trial [40] was the only study to report on other outcomes (Table 2). The criteria used to diagnose diarrhea were, however, only defined in one of the included studies [15,39]. This was a well-designed case-control study [15] reporting a similar risk of PKV detection in ileal material in neonatal (3–7-day old) piglets developing diarrhea (93.6%, *n* = 47) and age-matched controls staying non-diarrheic until inclusion (89.8%, *n* = 49). The use of quantitative real-time PCR made it possible to compare viral loads (log10 copies per reaction) in the positive pigs, but this also showed similarity between groups; the mean viral load was 4.60 (SD ± 1.76) in the PKV cases and 4.79 (SD ± 1.72) in the controls. The researchers concluded that PKV is most likely not a primary pathogen in the new neonatal diarrhea syndrome in pigs [15]. Their conclusion seems well-grounded in the presented evidence.

The objective of a Belgian project was to investigate the potential of nanopore sequencing technology in diagnostics for porcine health management [14]. The paper included a small cohort study of five pigs in a commercial farrowing unit. The pigs were followed from birth until 22 days of age. At eight time points during the study period, signs of diarrhea were registered, and rectal swabs were collected and subjected to RT-PCR for PKV as well as Rotavirus A and C. All piglets started to shed PKV during the first week of life, and the peak levels of shedding (between ~4.4 and 7.01 log10 copies/swab) were typically reached in the second week of life. A high shedding level persisted in two pigs, but this was not associated with diarrhea. Diarrhea was observed in two out of five piglets at day 5 and at days 5 and 8, respectively, and these episodes did not coincide with high levels of PKV shedding. One piglet died shortly after the peak in PKV shedding at day 11 [14]. Due to the small sample size (*n* = 5), this study provides little clarity on the intestinal pathogenicity of PKV.

The final study included in the present review was an experimental trial in which five colostrum-deprived newborn piglets were inoculated by mixing filtered fecal material into the milk they were supplied. The fecal material was collected from a diagnostic submission from a diarrhea outbreak among suckling piglets in a commercial farm, and it was infected with both PKV and porcine epidemic diarrhea virus. For comparison, five colostrum-deprived littermates were provided milk without fecal material. Viral shedding was measured by RT-PCR. The pigs receiving fecal material all developed severe disease, displaying diarrhea and vomiting from 12 h post inoculation, and they all died within 6 days after inoculation. The pigs receiving only milk stayed healthy. While the authors concluded that this indicated a possible role for both PKV and porcine epidemic diarrhea virus as gastrointestinal pathogens [40], some evidence points towards porcine epidemic diarrhea virus being responsible for the severe clinical outcomes. As the viruses were provided in a cocktail together, the effect of either virus cannot be separated. Yet, porcine epidemic diarrhea virus is well established as being very pathogenic [51], and thus severe disease would be expected irrespective of the presence of PKV in the fecal inoculum. Therefore, the study cannot provide good evidence of PKV being a gastrointestinal pathogen. Furthermore, timeliness has been discussed as a criterion to evaluate when establishing evidence of causation. It has been argued that X occurring before Y does not necessarily imply that X causes Y; however, if X occurs after Y, X may not be the cause of Y [50]. In this light, the daily measurements of viral load are interesting. Onset of clinical disease was observed already 12 h after inoculation [40]. At the first measurement of viral loads, 24 h after inoculation, the PKV shedding was rather modest (mean ≈ 1.8 log copies/mL). PKV did not reach a relatively high level (mean ≈ 4.2 log copies/mL) until the next measurement, two days after inoculation [40]. On the other hand, the porcine epidemic diarrhea virus shedding rapidly increased to its peak level (mean ≈ 5.8 log copies/mL) at the first measurement [40], thus appearing concurrently with the onset of disease. Along with prior knowledge, this timing of events further strengthens the suspicion of porcine epidemic diarrhea virus being the primary cause of disease, and it is less compatible with PKV being the primary cause of disease in this experimental trial. Nevertheless, it remains unknown whether the pigs would have gotten sick if they had only been given PKV.

### 4.3. Is Porcine Kobuvirus Causing Gastroinstestinal Disease in Young Pigs?

The present paper set out to review the evidence, possibly providing clarity to the question of whether PKV is a cause of gastrointestinal disease. In their paper from 2013, Verma and colleagues hypothesized four possible roles of PKV in this regard: *“(i) there exist two different pathotypes of porcine kobuvirus, namely, pathogenic and non-pathogenic; (ii) the virus load is higher in sick pigs versus healthy ones; (iii) kobuvirus causes diarrhea only in the presence of other pathogens, such as rotavirus or other enteric viruses; and (iv) kobuvirus is an endogenous passenger virus and is of no consequence“* [48]. The third hypothesis may deserve the addition that not only infectious causes may reduce the resilience of pigs to an extent where an otherwise harmless microbe may produce disease. Apart from this, the four hypotheses offer simple and useful, yet appropriate, categorizations capturing the nature of most viruses detected in pigs.

Summing up the evidence reviewed in the present review, 15 years of PKV research has not brought the field much closer to clarifying the four hypotheses. The present review only identified a single high-quality study, a case-control study, providing evidence indicating that PKV was not a cause of neonatal diarrhea in Danish pig productions [15]. The included cohort study suffered from a very small sample size, and in the experimental trial, the effect of PKV inoculation could not be separated from the effect of being inoculated with porcine epidemic diarrhea virus. In addition to these studies, more than 4000 young pigs had been assigned a diarrheic status and had their feces analyzed for PKV (Table 3). Unfortunately, little attention has been paid to the art of obtaining well-characterized, unbiased random samples, and therefore little inference can be drawn from these studies. That said, the data (Table 3) may indicate overall that a very strong association between PKV and diarrhea is less likely. The fact that PKV was commonly detected in non-diarrheic pigs (Table 3) is not evidence that it is not a cause of gastrointestinal disease. It could, however, indicate that PKV is not a sufficient cause (see [52] defining this term) in itself, and thus, if PKV is pathogenic, it likely only constitutes a minor component of a sufficient cause. It may also be explained by the hypothesis that reinfections of individuals with some immunological protection due to previous infections are common. This phenomenon is known in humans, where rotavirus infection often causes diarrhea in infants, but mainly in primary infections. The incidence rate of succeeding infections is high, but these infections are less commonly associated with diarrhea, and therefore rotavirus may often be isolated from non-diarrheic individuals who have previously experienced rotavirus as a cause of diarrhea [53].

Conclusively, there is a lack of good evidence for PKV being a cause of gastrointestinal disease, but the sparse available evidence supports the hypothesis that PKV is of limited clinical importance.

### 4.4. Recommendations for Future Research

As concluded in the preceding paragraph, there is a lack of evidence for PKV being a cause of diarrhea in young pigs. It is the opinion of the author of the present review that future research on PKV should prioritize establishing better evidence regarding the clinical importance of PKV before conducting more research on topics such as the genetics or the spatiotemporal distribution of PKV. What good is vast knowledge about a virus that might be clinically irrelevant?

As reviewed in the introduction, the prevalence of PKV has been demonstrated in pig production in numerous countries around the world (Oceania being an exemption). Therefore, studies solely describing the detection of PKV in a new region should not be viewed as significant contributions to the field.

The present review identified numerous poorly defined observational studies (Table 3). While some of them acknowledged the limitations implied by the sample collection at hand, others proceeded and made statistical tests or models of the association between PKV and diarrhea. Future observational studies should adopt classical epidemiological designs, such as the cross-sectional, cohort, or case-control design, allowing for stronger inferences. Studies without planned sampling strategies should not apply inappropriate statistical analyses.

In production herds, diarrhea in young pigs is typically a multifactorial problem. Some of the studies presented in Table 3 [9,11,23,26,35,37,47,49] as well as the cohort [14] and the case-control study [15,39] investigated multiple pathogens and found that PKV often occurred as a co-infection with known porcine pathogens, e.g., rotavirus or porcine epidemic diarrhea virus. For instance, Park and colleagues analyzed their collected samples for a broad variety of pathogens known to cause diarrhea in pigs. The vast majority (95.8%, *n* = 68) of the PKV-infected pigs with diarrhea were co-infected with known intestinal pathogens, even though PKV was the sole pathogen detected in three pigs with diarrhea [23]. This exemplifies why co-infections are important to consider as confounding factors, and therefore, it will often require multivariable statistical models to disentangle the effect of PKV on gastrointestinal disease outcomes. If ignoring co-infections, associations between PKV and gastrointestinal health outcomes (e.g., diarrhea) may not represent causal effects but arise solely due to confounding co-infections. The principle is exemplified in a directed acyclic graph in Figure 1, using rotavirus as an example. Rotavirus A is a well-established diarrheal pathogen in young pigs [54,55,56]. The occurrence of both PKV and rotavirus A typically peaks shortly after weaning [35,57,58] and co-infections with the two viruses were commonly observed in suckling pigs in poorly defined observational studies (e.g., [26,41]). Accordingly, rotavirus and PKV likely share transmission patterns; the type of contact in which transmission occurs is the same for the two viruses (getting in contact with fecal material from an infected pig); and the factors regulating the susceptibility to the viral infections (e.g., maternal immunity [59] or weaning-induced stress [60,61]) may also be shared for the two viruses. If these reasonable assumptions are true, it will lead to an association between rotavirus infection and PKV infection within individual pigs, even though there is no causal relation between being infected with the two viruses (i.e., PKV does not cause rotavirus or vice versa). Following this line of thought, PKV may simply be a proxy measure of other pathogens, such as rotavirus A. This can only be realized if studies consider multiple pathogens. The minimal set(s) of variables to be adjusted for may be identified by applying causal inference theory to the directed acyclic graph in Figure 1 [62]. In order to estimate the causal effect of PKV on diarrhea, two different sets of variables should be adjusted for: either the rate of contacts encountered and susceptibility to viral infections should be adjusted for, or, alternatively, one may adjust for rotavirus infection. As the latter variable will typically be much easier to measure, it appears to be the best choice in many settings. It should be noted that Figure 1 is a simplified example; multiple common pathogens may be relevant to consider.

Good evidence clarifying the pathogenicity of PKV may also be obtained from experimental trials inoculating pigs with PKV and mock-inoculating controls. However, PKV has been difficult for researchers to isolate and amplify, thus limiting the possibility of carrying out such studies. Thus, an objective for future research may be isolating and growing PKV in cell culture.

### 4.5. Recommendations for Pig Practitioners

In the light of the conclusions of this review, what should veterinary pig practitioners and diagnostic laboratories do? The author recommends not including PKV in differential diagnostic considerations for gastrointestinal disease problems or in standard microbiological laboratory analyses offered by veterinary diagnostic laboratories. This notion is not based on a certainty that PKV is not a cause of gastrointestinal disease; the present review concludes we do not know that and further evidence is warranted. The notion is based on the pragmatic viewpoint that, until further evidence has been established, the veterinarian is left with little idea on how to interpret tests positive for PKV. Therefore, detection of PKV in samples from diseased pigs will not help the veterinarian make better clinical decisions. Tests for PKV may be considered if profound diagnostic investigations (see [64]) have ruled out all other common causes of gastrointestinal disease.

## 5. Conclusions

Conclusively, there is a lack of good evidence that PKV is a cause of gastrointestinal disease, although the sparse available evidence suggests that PKV is of limited clinical importance. Future studies on PKV should prioritize clarifying this question by adopting high-quality study designs. Until then, veterinary pig practitioners should not be too concerned about PKV.

## Figures and Tables

**Figure 1 vetsci-10-00286-f001:**
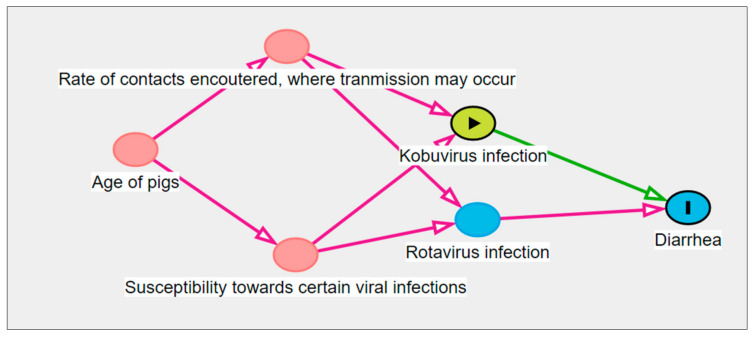
Directed acyclic graph proposing variables possibly confounding the causal effect of porcine kobuvirus on diarrhea incidence in weaned pigs. The figure was created by the author using http://www.dagitty.net/dags.html [63]. Kobuvirus infection may be associated with rotavirus infection (and other viral infections) because they have shared ancestors (i.e., they are both effects of the same cause). In epidemiological studies, this confounding will introduce bias to the estimate of the causal effect of the exposure, kobuvirus infection, on the outcome, diarrhea, if appropriate adjusting is ignored. Using causal inference theory, the minimal set of variables to be adjusted for to identify the causal effect of interest is either susceptibility to certain viral infections and the rate of contacts encountered, where transmission may occur, or Rotavirus infection. The latter will often be the easiest to record.

**Table 1 vetsci-10-00286-t001:** Specification of the four items defining the clinical question answered in this review.

Item	Definition
Population	Pigs less than 22 weeks of age living in commercial, intensified indoor pig productions.
Intervention/exposure	Infection with porcine kobuvirus (PKV): inoculation or PKV detected with any diagnostic test in any sample material, including tissue, blood, and feces.
Comparison	No infection with porcine kobuvirus (PKV): not inoculated or PKV was not detected with any diagnostic test in any sample material, including tissue, blood, and feces; for quantitative diagnostics, PKV detected at lower levels in any sample material, including tissue, blood, and feces.
Outcome	Gastrointestinal disease: clinical signs, e.g., altered fecal consistency or color, fecal deposits such as blood or mucus, vomiting, anorexia, etc. Biochemical signs of gastrointestinal disease, e.g., blood electrolyte imbalances or alteration of biomarkers in the blood. Macroscopic or microscopic lesions present in the gastro-intestinal tract.

**Table 3 vetsci-10-00286-t003:** Occurrence (*n* and %) of porcine kobuvirus (PKV) in diarrheic and non-diarrheic pigs in poorly defined observational studies.

Reference	Age (Weeks)	Diarrhea		Not Diarrhea		Diarrhea	Not Diarrhea
		PKV+	PKV−	PKV+	PKV−	PKV Risk	PKV Risk
[41]	*	28	58	15	18	32.6%	45.5%
[42]	<4	52	16	10	15	76.5%	40.0%
	4 < 7	7	8	2	6	46.7%	25%
	7 < 23	3	7	1	6	30%	14.3%
[43]	“Piglets”	505	23	104	4	95.6%	96.3%
[44]	<6	34	61	0	40	35.8%	0%
[11,35] †	<1	9	30	4	8	23.1%	33.3%
	<1	†	†	9	31	†	22.5%
	1 < 3	20	2	8	3	90.9%	72.7%
	1 < 3	†	†	27	30	†	47.4%
[26]	<4	44	15	56	31	74.6%	64.4%
	4 < 10	41	32	50	24	56.2%	67.6%
	>10	19	12	52	38	61.3%	57.8%
[23]	<3	45	1	6	9	97.8%	40.0%
	3 < 6	15	3	4	11	83.3%	26.7%
	6 < 22	7	4	4	25	44.4%	13.8%
[47]	<4	1	3	4	3	25%	57.1%
	4 < 10	3	2	0	4	60%	0%
[45]	0-“weaners”	59	180	49	96	24.7%	33.8%
[29]	<7	7	4	46	32	63.6%	59%
	7 < 14	6	26	38	109	18.8%	25.9%
	15 < 23	2	6	19	98	25%	16.2%
[48]	‡	25	89	10	36	21.9%	21.7%
[46]	#	30	19	15	52	61.2%	22.4%
[49]	1	41	540	23	92	7.1%	20%
	1 < 3	10	118	15	37	7.8%	28.9%
	3 < 10	24	75	24	23	24.2%	51.1%
	10 < 24	6	70	19	33	7.9%	36.5%
[9]	0 < 5	57	29	37	25	66.3%	59.7%
	5 < 10	12	7	22	25	63.2%	46.8%
	11 < 18	21	17	2	2	55.3%	50%

* The 60 pigs aged less than 3 weeks and the 59 “growing-finishing pigs” [41]. † The numbers are approximations from a bar chart showing percentages. Two non-diarrheic comparison groups were sampled: non-diarrheic pigs sharing a pen with at least one diarrheic pig (upper row) and non-diarrheic pigs from pens with no diarrhea (lower row) [11]. ‡ The study included the groups “less than 4 weeks”, “less than 8 weeks”, and “pigs older than 9 weeks” [48]. # From birth (“less than 6 weeks”) and finisher pigs older than 12 weeks of age [46].

## Data Availability

Not applicable.

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
