# Peer review of "A Systematic Review: Is Porcine Kobuvirus Causing Gastrointestinal Disease in Young Pigs?"

_vetsci, 2023, doi:10.3390/vetsci10040286_

Round 1

Reviewer 1 Report

The author in his systematic review aimed to clarify the role of porcine kobuvirus as a potential cause of gastrointestinal disease in piglets. From 110 relevant publications only 17 were included in the review as the rest did not meet the PICO-based criteria. The author thoroughly reviewed all included publications as to the study type, age of animals and outcome of infection exposure. All those studies were discussed in details and the conclusion of the review together with recommendations for pig practitioners were made.

The manuscript is well and comprehensibly written, however, the main question about the role of PKV as gastrointestinal pathogen in pigs have not been answered satisfactorily yet. I have just several minor comments mostly concerning the English.

P1, L28: “… this could point towards PKV is not a sufficient cause in itself …”. Please, rephrase this sentence (“point toward” is used with a noun, so the sentence should be e.g.: “… this could point toward the fact that PKV is not…”. Or another wording is possible, e.g.: “… this could indicate that PKV is not…”). Please, correct the same mistake also on P8, L299.

P1, L40: Kobuvirus has been detected also in sheep.

P2, L57: “The pathogenic capabilities of PKV was questioned …” Please, correct the mistake (“The pathogenic capabilities of PKV were questioned …”).

P2, Table 1: “… i.e. PKV not inoculated, or PKV detected with …” The sentence should correctly be “… i.e. PKV not inoculated, or PKV not detected with …”. And further “… PKV detected lower levels …” should be “… PKV detected in lower levels …”.

P3, L90-91: “Data was extracted form a set of observational studies, as far possible separated into age groups, …” The sentence does not make a good sense, please rephrase. (e.g., “Data were extracted from a set of observational studies; if possible, separated into age groups, …”).

P5, L144: “… subparagraphs offers …”. Correctly it is “… subparagraphs offer …”.

P6, L153: “… from 15 different states farms …”. Did the author mean that the samples were from 15 state farms or that the farms were from 15 different states? If the second possibility is the case, then the sentence should be e.g. “… from farms located in 15 different states …”.

P6, L156: “Weaned pigs was …”. Please, correct the typo (“Weaned pigs were …”).

P6, L181: “… matching have impossibly been applied.” The sentence would be easier to understand as e.g. “… matching could not be applied.”

P6, L184: “… obtainment of an unbiased, representative samples …”. “Samples” is plural, so correct the sentence (“… obtainment of unbiased, representative samples …”)

P6, L197: “… occurrence PKV detection in diarrheic …” This seems too complicated. “… PKV detection in diarrheic …” should be enough.

P7, L213: “… readers should be acquainted that many …” Please, re-word the sentence (e.g., “… readers should be aware that many …”).

P8, L288-289: “… study providing evidence indicating that PKV providing some evidence pointing towards PKV not being a cause of neonatal diarrhea Danish pig productions.” This sentence contains too many words. Better version could be e.g., “… study providing evidence indicating that PKV is not a cause of neonatal diarrhea in Danish pig productions.”

P8, L292: “… the effect of being inoculated porcine epidemic diarrhea virus.” This sentence should be corrected (e.g. “… the effect of inoculation with porcine epidemic diarrhea virus.”)

P8, L299: “… cause of gastrointestinal.”. There is a missing word (“… cause of gastrointestinal disease.”)

P9, L305: “The incidence rate of succeeding infections are high, …”. “Rate” is singular, so the correct sentence is “The incidence rate of succeeding infections is high,…”

P9, L309: “… the sparse available evidence supported that PKV …”. Please, rephrase (e.g., “… the sparse available evidence supported the assumption that PKV …”, or “… the sparse available evidence suggests that PKV …”)

P9, L313: “It is the opinion of the author the present review that future research in PKV …”. Please, rephrase (e.g., “It is the opinion of the present review author that future research on PKV …”).

P9, L336: “… even though PKV were the sole pathogen …”. PKV is singular, please, correct (“… even though PKV was the sole pathogen …”).

P9, L354: “… PKV may simple be …”. Correctly it is “… PKV may simply be …”.

P10, L367: “… may be associated to rotavirus infection …” Correct, please (“… may be associated with rotavirus infection …”).

P10, L371: “… set of variable …” Please, correct “… set of variables …”.

Author Response

I would like to thank the reviewer for reviewing my paper. I am truly grateful for the careful revision of grammatical deficiencies and typos.

P1, L28: “… this could point towards PKV is not a sufficient cause in itself …”. Please, rephrase this sentence (“point toward” is used with a noun, so the sentence should be e.g.: “… this could point toward the fact that PKV is not…”. Or another wording is possible, e.g.: “… this could indicate that PKV is not…”). Please, correct the same mistake also on P8, L299.

This has been corrected:

”this could indicate that PKV” (L28)

“It could, however, indicate that PKV is” (L299)

P1, L40: Kobuvirus has been detected also in sheep.

The sentence has now been revised, and a reference to the first detection in sheep has been added:  “Since then, kobuviruses have been discovered in domestic animals including cats [3], cattle [4], dogs [5], goats [6], sheep [7], and pigs.”

P2, L57: “The pathogenic capabilities of PKV was questioned …” Please, correct the mistake (“The pathogenic capabilities of PKV were questioned …”).

This has revision has been implemented exactly as suggested.

P2, Table 1: “… i.e. PKV not inoculated, or PKV detected with …” The sentence should correctly be “… i.e. PKV not inoculated, or PKV not detected with …”. And further “… PKV detected lower levels …” should be “… PKV detected in lower levels …”.

These has revisions have been implemented exactly as suggested.

P3, L90-91: “Data was extracted form a set of observational studies, as far possible separated into age groups, …” The sentence does not make a good sense, please rephrase. (e.g., “Data were extracted from a set of observational studies; if possible, separated into age groups, …”).

The sentence has been rephrased and separated into two, now reading: “Data was extracted form a set of observational studies (see Table 3). Whenever possible, the data was collected stratified by age groups, (…)”

P5, L144: “… subparagraphs offers …”. Correctly it is “… subparagraphs offer …”.

This has revision has been implemented exactly as suggested.

P6, L153: “… from 15 different states farms …”. Did the author mean that the samples were from 15 state farms or that the farms were from 15 different states? If the second possibility is the case, then the sentence should be e.g. “… from farms located in 15 different states …”.

Yes, this was a typo. The sentence has now been rephrased:

“For instance, in an American study, diarrheic samples submitted to a laboratory from farms located in 15 different states were compared to non-diarrheic samples collected from three farms in Minnesota [48]. “

P6, L156: “Weaned pigs was …”. Please, correct the typo (“Weaned pigs were …”).

This has revision has been implemented exactly as suggested

P6, L181: “… matching have impossibly been applied.” The sentence would be easier to understand as e.g. “… matching could not be applied.”

This has been rephrased to: “matching have not been applied”

P6, L184: “… obtainment of an unbiased, representative samples …”. “Samples” is plural, so correct the sentence (“… obtainment of unbiased, representative samples …”)

This has revision has been implemented exactly as suggested: “(…) the obtainment of unbiased, representative samples may (…)”

P6, L197: “… occurrence PKV detection in diarrheic …” This seems too complicated. “… PKV detection in diarrheic …” should be enough.

This has been rephrased: “(PKV occurrence in diarrheic and non-diarrheic pigs)”

P7, L213: “… readers should be acquainted that many …” Please, re-word the sentence (e.g., “… readers should be aware that many …”).

This has been rephrased: “(…) readers should be aware that many (…)”

P8, L288-289: “… study providing evidence indicating that PKV providing some evidence pointing towards PKV not being a cause of neonatal diarrhea Danish pig productions.” This sentence contains too many words. Better version could be e.g., “… study providing evidence indicating that PKV is not a cause of neonatal diarrhea in Danish pig productions.”

This has been rephrased: “The present review only identified a single high-quality study, a case-control study, providing evidence indicating that PKV is not a cause of neonatal diarrhea in Danish pig productions [15].”

P8, L292: “… the effect of being inoculated porcine epidemic diarrhea virus.” This sentence should be corrected (e.g. “… the effect of inoculation with porcine epidemic diarrhea virus.”)

This has revision has been implemented exactly as suggested: “(…)of being inoculated with porcine epidemic diarrhea virus”

P8, L299: “… cause of gastrointestinal.”. There is a missing word (“… cause of gastrointestinal disease.”)

Yes, this was a missing word. It now reads: “(…) cause of gastrointestinal disease.”

P9, L305: “The incidence rate of succeeding infections are high, …”. “Rate” is singular, so the correct sentence is “The incidence rate of succeeding infections is high,…”

This has revision has been implemented exactly as suggested: “The incidence rate of succeeding infections is high (…)”

P9, L309: “… the sparse available evidence supported that PKV …”. Please, rephrase (e.g., “… the sparse available evidence supported the assumption that PKV …”, or “… the sparse available evidence suggests that PKV …”)

This has been rephrased to read: “ Conclusively, there is a lack of good evidence of PKV being a cause of gastrointestinal disease, but the sparse available evidence supports the hypothesis that PKV is of limited clinical importance.”

P9, L313: “It is the opinion of the author the present review that future research in PKV …”. Please, rephrase (e.g., “It is the opinion of the present review author that future research on PKV …”).

This has been rephrased to read: “It is the opinion of the author of the present review that future research on PKV (…)”

P9, L336: “… even though PKV were the sole pathogen …”. PKV is singular, please, correct (“… even though PKV was the sole pathogen …”).

This has revision has been implemented exactly as suggested.

P9, L354: “… PKV may simple be …”. Correctly it is “… PKV may simply be …”.

This has revision has been implemented exactly as suggested.

P10, L367: “… may be associated to rotavirus infection …” Correct, please (“… may be associated with rotavirus infection …”).

This has revision has been implemented exactly as suggested.

P10, L371: “… set of variable …” Please, correct “… set of variables …”.

This has revision has been implemented exactly as suggested.

Reviewer 2 Report

The submitted manuscript is of the review type and is prepared at a very good level. I believe that its focus is suitable for your magazine. I only have small comments about it.

Keywords are repeated in the title.

The author states that the virus was found in other types of domestic animals. I know that rewiu applies only to pigs, but I think it would be appropriate to briefly state how serious the infection caused by this virus is in other types of animals.

I assume that figure 1 was made by the author himself. I recommend stating this fact in the description of the picture.

Author Response

Thank you for reviewing my paper.

Keywords are repeated in the title.

Response: I’m uncertain about his comment. Does the reviewer mean that keywords should preferably not repeat words which are already included in the title? To my knowledge keywords are used for indexing by librarians and I am not fond of omitting the words pig, kobuvirus, or gastrointestinal disease from neither the title nor the keywords.

The author states that the virus was found in other types of domestic animals. I know that rewiu applies only to pigs, but I think it would be appropriate to briefly state how serious the infection caused by this virus is in other types of animals.

Response: Based on the experience gained by conducting the present review, I do not feel comfortable in providing such brief statements without having conducted a systematic review, similar to the present, for the other domestic species. Picking out few references may easily be misleading. I could identify a recent review (https://doi.org/10.3390/microbiolres 12030048) briefly discussing the issue in cattle. However, I not do not feel comfortable in judging the comprehensiveness of this narrative review, and hence I will not dare to recommend it and its conclusions. Furthermore, they actually conclude that the pathogenicity appears unclear at this stage. Another review had the objective to “review taxonomy and genome organization, detection of BKV in different countries and genomic characterization” (https://doi.org/10.1111/tbed.13909 ). The authors proceed beyond this objective and briefly discuss the pathogenicity. Despite presenting a body of evidence even poorer than the studies listed in Table 3 of the present paper, they conclude: “Our and other studies from South Korea (Jeoung et al., 2011), Turkey (Isidan et al., 2019), Egypt (Mohamed et al., 2018) and Brazil (Ribeiro et al., 2014) indicate that BKV might be an infectious agent involved in neonatal calf diarrhoea.” I must admit, that I find their review of the included observational studies uncritical, and I am not convinced. Conclusively, I am not confident in making any brief statements for any domestic species with the literature at hand, and this may indicate the need for work similar to the present in other domestic species.

I assume that figure 1 was made by the author himself. I recommend stating this fact in the description of the picture.

Response: The first line in the figure description now reads:  

“The figure was created by the author of this review using http://www.dagitty.net/dags.html [63].”

Thus, it now clearly states that this is an original figure made by the author, and additionally a reference “[63]” describing the tool, dagitty, has been added.

Reviewer 3 Report

This is a well-written paper regarding kobuvirus in pigs (PKV). A case-control study showed that PKV was not associated to neonatal diarrhea. The authors conclude that there is a lack of good evidence of PKV being a cause of gastrointestinal disease, but the sparse available evidence supported that PKV is of limited clinical importance.

Author Response

I thank the reviewer for the positive response.